# The photospheric origin of the Yonetoku relation in gamma-ray bursts

Hirotaka Ito[1,2], Jin Matsumoto [3], Shigehiro Nagataki[1,2], Donald C. Warren [1,3], Maxim V. Barkov[1,4] & Daisuke Yonetoku[5]

Long duration gamma-ray bursts (GRBs), the brightest events since the Big Bang itself, are believed to originate in an ultra-relativistic jet breaking out from a massive stellar envelope. Despite decades of study, there is still no consensus on their emission mechanism. One unresolved question is the origin of the tight correlation between the spectral peak energy and peak luminosity discovered in observations. This Yonetoku relation is the tightest correlation found in the properties of the prompt phase of GRB emission, providing the best diagnostic for the radiation mechanism. Here we present three-dimensional hydrodynamical simulations, and post-process radiation transfer calculations, of photospheric emission from a relativistic jet. Our simulations reproduce the Yonetoku relation as a natural consequence of viewing angle. Although jet dynamics depend sensitively on luminosity, the correlation holds regardless. This result strongly suggests that photospheric emission is the dominant component in the prompt phase of GRBs.

[1] Astrophysical Big Bang Laboratory, RIKEN, Saitama 351-0198, Japan. [2] Interdisciplinary Theoretical & Mathematical Science Program (iTHEMS), RIKEN, Saitama 351-0198, Japan. [3] School of Mathematics, Faculty of Mathematics and Physical Sciences, University of Leeds, Leeds LS2 9JT, UK. [4] Department of Physics and Astronomy, Purdue University, 525 Northwestern Avenue, West Lafayette, IN 47907-2036, USA. [5] College of Science and Engineering, School of Mathematics and Physics, Kanazawa University, Kakuma, Kanazawa, Ishikawa 920-1192, Japan. Correspondence and requests for materials should be addressed to H.I. (email: hirotaka.ito@riken.jp)

So far, no theoretical work has provided a fully consistent explanation for the origin of the Yonetoku relation[1,2]. Both the well-studied internal shock model[3] and the more recent magnetic reconnection model[4] lack the ability to make firm predictions about the resulting emission properties, since they invoke non-thermal plasma physics with large uncertainties. Too many parameters (e.g., particle acceleration efficiency and magnetization) do not have a strong constraint but must be specified to evaluate the non-thermal emission. As a result, in order to reproduce the observed correlation, one needs to assume that there is self-regulation among the imposed parameters[5]. However, it is not obvious why such self-regulation should occur across bursts.

In addition, models that invoke optically thin synchrotron emission also face problems in reproducing the spectrum (hard spectral slopes[6] and sharp spectral peak[7]) in a non-negligible fraction of GRBs. These problems arise from the fundamental physics of synchrotron emission and so cannot be explained within this framework.

The above difficulties have led recent theoretical and observational studies to consider photospheric emission (photons released from a relativistic jet during the transition from optically thick to thin states) as a promising alternative scenario[8–26]. This model predicts the emergence of quasi-thermal radiation and can reproduce those spectral shapes that are incompatible with synchrotron theory.

Another strong advantage of the photospheric model is that it does not require a large number of uncertain parameters, since it is based on thermal processes. Indeed, many studies have discussed the origin of the relation based on photospheric emission. However, these analyses adopted oversimplified jet dynamics (e.g., steady spherical flow)[11,12] and/or crude assumptions for radiation processes[13,14]. More sophisticated study is necessary to firmly connect photospheric emission to the Yonetoku relation. We do so, robustly, here.

For an accurate analysis of photospheric emission, the jet evolution and accompanying photons must be followed from their origin, deep within the star, to the point where photons fully decouple from the jet. This requires both relativistic hydrodynamics and full radiation transfer. To capture all the essential features, the calculation needs to cover a large range in time and space, and must be performed in three dimensions (3D). We have previously reported on such a calculation[15], which was followed by another group[16–18] in 2D. However, these studies were only able to evaluate the emission at small viewing angles $\theta_{obs}$. High latitude ($\theta_{obs} \gtrsim 4°$) emission lacked accuracy since the calculation domains ($\lesssim 10^{13}$ cm) were not sufficient for the photons to decouple from the fluid in the jet. Moreover, the studies explored only a small part of the parameter space, so it was unclear how emission depends on the intrinsic properties of the jet.

To examine these issues, we perform large scale 3D relativistic hydrodynamical simulations of jets breaking out of a massive stellar envelope[27], followed by a post-process radiation transfer calculation in 3D. This procedure is well tested[15], but to achieve full decoupling of photons from the jet we extend the calculation domain by a factor of ~20 in space and time compared to our previous study. Moreover, we perform three sets of simulations to cover a wide range of model parameters. In each simulation, a jet with a different kinetic power is considered: $L_j = 10^{49}$, $10^{50}$, and $10^{51}$ erg s$^{-1}$ (Methods section). We show that the Yonetoku relation is reproduced as a natural consequence of viewing effect regardless of the jet power.

## Results

**Hydrodynamical simulation.** Figure 1 shows an image of our hydrodynamical simulation for the $L_j = 10^{50}$ erg s$^{-1}$ model. Interaction with the stellar envelope, and the resultant formation of collimation shocks, most strongly influence the jet dynamics. Although this qualitative feature is common among the three models, it is most pronounced in the model with lowest jet power, since higher-power jets can penetrate the stellar envelope with less interaction. As a result, wobbling and complex structures are found throughout the outflow for the $10^{49}$ erg s$^{-1}$ model. In the other two cases, only the portion nearest the jet head shows such features; the jet maintains a steady laminar structure below.

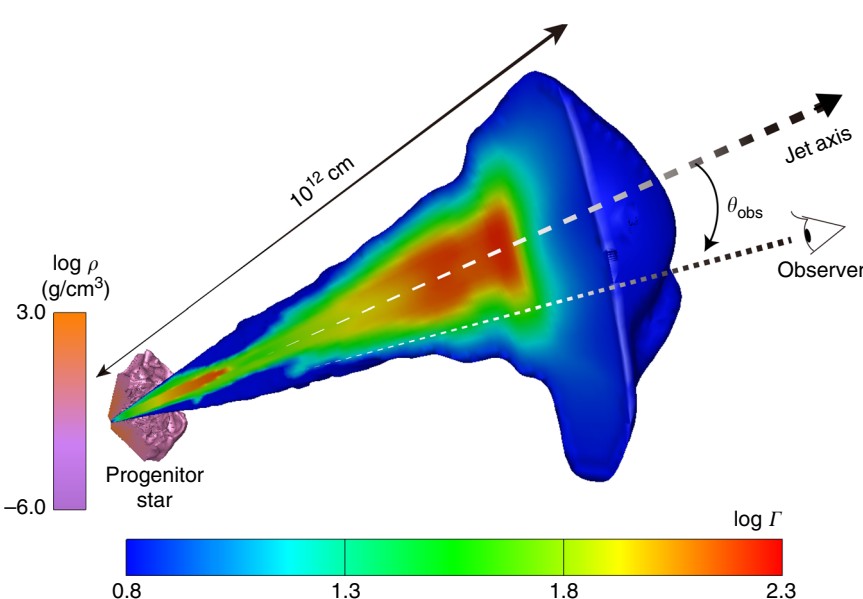

**Fig. 1** Snapshot of the hydrodynamical simulation. 3D profile with a 2D slice taken through the midplane of the simulation at a laboratory time $t = 40$ s for the model with jet power $L_j = 10^{50}$ erg s$^{-1}$. The profiles of the progenitor star and jet are visualized using color contours of mass density and Lorentz factor, respectively. Together with the simulation result, we also show the location of the jet axis (dashed arrow) and how we define the viewing angle $\theta_{obs}$ of an observer's line-of-sight (dotted line)

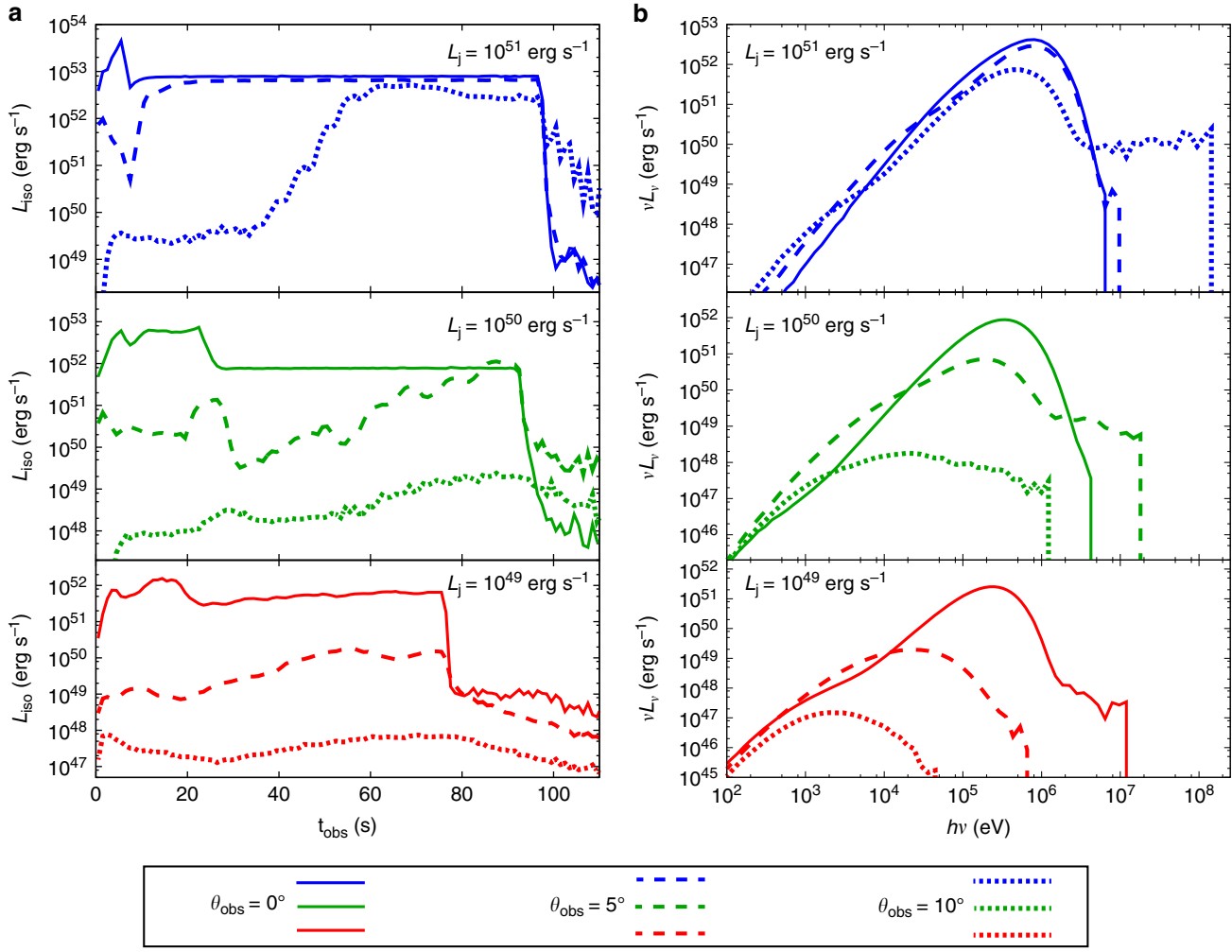

**Fig. 2** Light curves and time integrated spectra. **a** Light curves up to observer time $t_{obs} = 110$ s. **b** Time integrated spectra constructed by averaging over the duration $t_{dur} = 110$ s. Line color indicates jet power, with red, green, and blue showing the cases of $L_j = 10^{49}$, $10^{50}$, and $10^{51}$ erg s$^{-1}$, respectively. For each model, we show results for three different viewing angles: $\theta_{obs} = 0°$ (solid line), 5° (dashed), and 10° (dotted). Note that, although high-energy photons suffer from low statistics, this does not affect the evaluation of the overall luminosity or the spectral peak energy

**Light curve and spectrum**. The resulting emission is summarized in Fig. 2. Models with higher jet power tend to show higher luminosity and spectral peak energy. This is mainly due to the larger energy budget for emission, and the higher overall temperature of the jet, as the jet power and Lorentz factor increase.

In the light curves, notable time variability arises due to the structure developed via jet-stellar interactions. For an observer with small viewing angle $\theta_{obs}$, steady emission is observed at later phases, since the inner region with laminar structure becomes visible. The fact that this feature is not observed in GRBs suggests that the central engines of these events are either not extremely luminous or not steadily luminous.

Regarding the spectra, we find non-thermal features compatible with observations, even though only thermal photons are injected in the current work. The broadening from a thermal spectrum at energies below and above the spectral peak is mainly caused by the multi-temperature and bulk Comptonization effects, respectively, which are induced by the global structure of the jet[12,19]. We note, however, that an accurate evaluation of the spectral shape requires higher spatial resolution[15,25]. Moreover, if present, non-thermal particles arising from internal dissipation[20–24] may also contribute to spectral broadening. Note that such dissipation does not affect the average energy of photons as long as the generated heat is smaller than the thermal

energy. In the present study, we focus on the overall properties, such as spectral peak energy $E_p$ and peak luminosity $L_p$, that are largely unaffected by such ingredients.

**Yonetoku relation**. A comparison of the Yonetoku relation with our results is shown in Fig. 3. We plot $E_p$ and $L_p$ sampled from the entire duration ~100 s (roughly comparable to the duration of jet injection), but we also include the cases where only emission up to a certain duration (20, 40, and 60 s) is considered. This is intended to mimic bursts originating from shorter jet activity, since long GRBs have diversity in their durations. Since the early phase of the emission is nearly identical to the entire emission arising from shorter jet injection[26], we consider this simple change justified.

The lateral structure of the jet, developed during propagation through the stellar envelope, leads to a strong dependence on the viewing angle. Since the region near the jet axis has the highest Lorentz factor and temperature, one expects higher luminosity and spectral peak energy at smaller $\theta_{obs}$. This sequence produces a continuous correlation between $E_p$ and $L_p$ that spans several orders of magnitude. Though the distributions of $E_p$ and $L_p$ are shifted to higher values as the jet power increases, we find very similar behavior in all three models despite the variety in dynamics. All models reproduce the Yonetoku relation

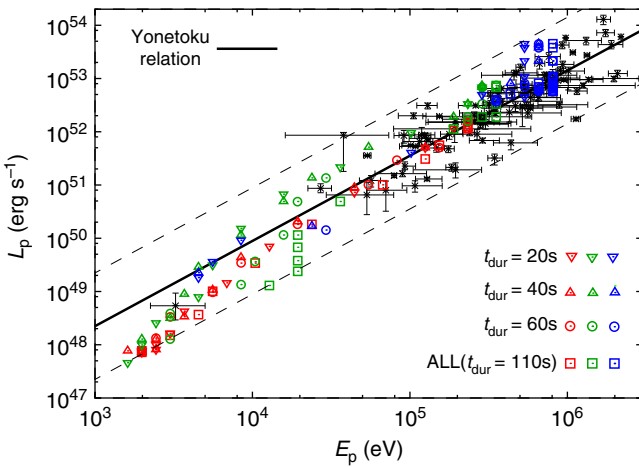

**Fig. 3** Relation between spectral peak energy $E_p$ and peak luminosity $L_p$. Results of our simulations are plotted with the observational data of 101 GRBs (gray points with error bars) and best-fit curve of the Yonetoku relation (black solid line), $L_p = 10^{52.43\pm0.037} \times [E_p/355\text{ keV}]^{1.60\pm0.082}$ erg s$^{-1}$, taken from the literature[2]. The error bars of the observational data indicate 1-$\sigma$ standard error for both $E_p$ and $L_p$. Two dashed lines located below and above the best-fit curve show the 3-$\sigma$ systematic error regions of the Yonetoku relation. Symbol color indicates model, with red, green, and blue representing jet powers of $L_j = 10^{49}$, $10^{50}$, and $10^{51}$ erg s$^{-1}$, respectively. The inverted triangle, triangle, circle, and square plot the results obtained by sampling the emission up to durations $t_{dur} = 20$, 40, 60, and 110 s, respectively. The considered range of viewing angle is $0° \leq \theta_{obs} \leq 11°$, with 1° between successive points. Although the current results do not extend up to the bright end of the observed distribution, we expect that this population can be naturally produced by increasing the jet power and/or Lorentz factor, which shift the population toward higher energy and luminosity in the present study

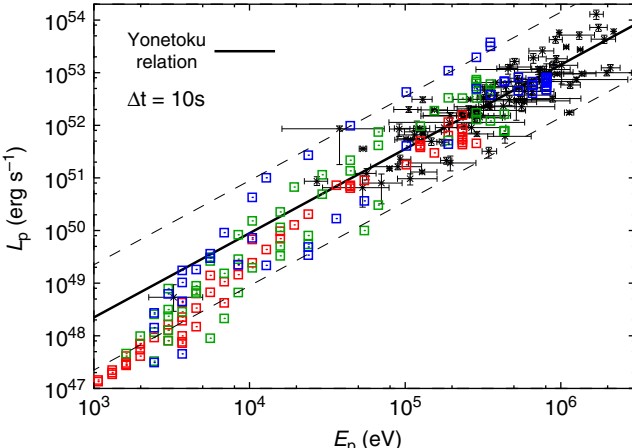

**Fig. 4** Relation between time-resolved spectral peak energy $E_p$ and peak luminosity $L_p$. Same as Fig. 3, but for a spectral peak energy and peak luminosity computed in five time intervals of $\Delta t = 10$ s successively taken within an observer time $t_{obs} = 50$ s. The error bars of the observational data indicate 1-$\sigma$ standard error for both $E_p$ and $L_p$

remarkably well. Since a wide range of jet power and duration is covered in our analysis, we stress that this is not the result of fine-tuning in our simulation setup but an inherent property of GRB photospheric emission.

We see some dependence on duration, as shorter durations tend to slightly shift the spectra to the softer side. This is because the region near the head of the jet is subject to baryon loading from the progenitor envelope, which pushes out the photosphere to larger distances, and therefore cools the radiation. Although this causes some dispersion in the correlation, all three models trace the Yonetoku relation regardless of duration. We conclude that the tight correlation is not affected by the duration of jet injection, or by the jet power.

## Discussion

Contrary to predictions in previous studies based on 1D models[11,12], the Yonetoku relation need not reflect diversity in the intrinsic properties of the jet. Instead, it naturally arises from the dependence of emission properties on the viewing angle. Bright, hard emission is observed on-axis, while soft, dim emission can be observed off-axis. Variation in the jet properties (e.g., power, Lorentz factor, and duration) appears as dispersion in $E_p$ and $L_p$ around the correlation curve, which also nicely reproduces the observed scatter.

Previous works that performed 2D hydrodynamical simulations[13,14] also claimed to reproduce the observed correlation between the spectral peak energy and the total radiated energy (the Amati relation)[28] through changing the viewing angle. However, their calculations did not include radiation transfer, which is essential for the evaluation of the emission.

Indeed, recent simulations that do incorporate radiation transfer calculation show deviations from previous results[16,17]. We note again that the imposition of 2D axisymmetry and the limited calculation domain can cause inaccuracy. The former assumption induces error in the evaluation of emission, particularly along the jet axis because of the coordinate singularity, while the latter ingredient prevents robust predictions of off-axis emission. Our current study overcomes both issues and shows that the Yonetoku relation is an inherent feature of photospheric emission in GRBs.

While our results show a continuous sequence over three orders of magnitude in $E_p$, the observational data are limited at luminosities below ~$10^{50}$ erg s$^{-1}$ due to difficulties detecting dim transients. We find excellent agreement at high luminosities (>$10^{50}$ erg s$^{-1}$) where the observations do not suffer from possible biases suggested in the literature[29]. Here the best-fit curve of our simulation is given by $L_p = 10^{52.6} \times [E_p/355\text{ keV}]^{1.67}$ erg s$^{-1}$, which is consistent with the observations (see Fig. 3). On the other hand, at low luminosities where observations do not provide a strong constraint, we find a slight deviation of population away from the best-fit curve of the Yonetoku relation toward higher peak energy $E_p$. Nevertheless, the simulation results also overlap with the only existing observational data at such low luminosities. This may be indicating that the correlation curve has a steeper slope at this luminosity range. We look forward to future observations falling in this part of the $E_p$–$L_p$ plane.

Although not as established as the correlation found among the bursts, there is an important indication in literature[30–32] that $E_p$–$L_p$ correlation also holds at any time interval of individual bursts. To see whether such tendency is also found in our calculation, we performed a time-resolved analysis of our results. Here, we have taken uniform time intervals of 10 s and determined the spectral peak and peak luminosity within each interval. The results are displayed in Fig. 4. As shown in the figure, we also find a good agreement with the correlation curve. Hence, our calculation supports the picture that $E_p$–$L_p$ relation is also satisfied within an individual burst.

## Methods

**Hydrodynamical simulation**. In order to evaluate the long-term evolution of a relativistic jet that penetrates a massive star, we have performed special relativistic hydrodynamical simulations in a three-dimensional spherical coordinate system $(r, \theta, \phi)$. The numerical code is identical to that of our previous work[15]. The main

difference is that we have followed the evolution for a longer time scale (up to $t = 6000$ s) and, therefore, larger spatial scale (up to $r \sim 2 \times 10^{14}$ cm). Note that this is mandatory to ensure that the photons are fully decoupled from the jet even at high latitudes[15,16]. In order to accomplish the calculation within a reasonable computational time, we have done six remappings in the radial direction. The corresponding time and radial spatial domain at each remapping are as follows:

| | |
|---|---|
| 0th | $0 \text{ s} \leq t \leq 130$ s |
| | $10^{10} \text{ cm} \leq r \leq 8 \times 10^{12}$ cm |
| 1st | $130 \text{ s} < t \leq 200$ s |
| | $10^{11} \text{ cm} \leq r \leq 2.9 \times 10^{13}$ cm |
| 2nd | $200 \text{ s} < t \leq 1000$ s |
| | $10^{12} \text{ cm} \leq r \leq 6.1 \times 10^{13}$ cm |
| 3rd | $1000 \text{ s} < t \leq 2000$ s |
| | $8 \times 10^{12} \text{ cm} \leq r \leq 1 \times 10^{14}$ cm |
| 4th | $2000 \text{ s} < t \leq 3500$ s |
| | $2.9 \times 10^{13} \text{ cm} \leq r \leq 1.4 \times 10^{14}$ cm |
| 5th | $3500 \text{ s} < t \leq 5000$ s |
| | $6.1 \times 10^{13} \text{ cm} \leq r \leq 1.8 \times 10^{14}$ cm |
| 6th | $5000 \text{ s} < t \leq 6000$ s |
| | $10^{14} \text{ cm} \leq r \leq 2.2 \times 10^{14}$ cm |

In each remapping process, we shift the position of the inner and outer boundaries to larger distances by discarding and adding inner and outer grid zones, respectively, so that the entire structure of the propagating jet is contained within the calculation domain at any time. The two angular coordinates are fixed at $\pi/4 \leq \theta, \phi \leq 3\pi/4$ in all remappings. We impose a reflective boundary condition on the initial (0th) inner boundary in the radial direction except for the jet injection region, while an outflow (zero gradient) boundary condition is employed after the 1st remapping. The outflow boundary condition is also used for the outer boundary of the radial grid as well as all four boundaries along the side of the grid throughout the calculation.

As for the spatial resolution of the calculation, 280 uniformly spaced grid zones are used for $\theta$ and $\phi$, while 1260 non-uniform grid zones are employed for the $r$ coordinate. The grid size in the radial direction increases with radius as $\Delta r = \Delta\theta\, r [1 + r/r_0]^{-1}$. In this equation, $\Delta\theta = \pi/560$ is the angular grid size and $r_0 = 2 \times 10^{13}$ cm is the reference position beyond which radial grid size asymptotically approaches a constant value. The suppression of increase in grid size at $r > r_0$ is introduced to maintain the radial resolution at a level where the overall jet structure can be resolved even at large radius, $r \sim 10^{14}$ cm.

The above resolution is too coarse to capture variability of emission much shorter than a second. Note, however, that such a short time scale is not crucial to the current study, since the Yonetoku relation is determined with time bins of a second.

In solving the hydrodynamics, we use a numerical code[33] which employs a relativistic HLLC Riemann solver scheme. A MUSCL-type interpolation method is used to attain second-order accuracy in space, with second-order temporal accuracy using Runge–Kutta time integration. We assume an ideal gas, $p = (\gamma - 1)\rho\varepsilon$, with $p$, $\gamma = 4/3$, $\rho$, and $\varepsilon$ being pressure, specific heat ratio, rest mass density, and specific internal energy, respectively.

Though they are irrelevant in the hydrodynamical evolution, the local temperature, $T$, and number density of electrons, $n_e$, must be specified for the calculation of radiation transfer. The temperature is determined from the pressure by assuming a radiation dominated gas, namely, $p = a_{rad}T^4/3$, where $a_{rad}$ is the radiation constant. The electron number density is determined from the mass density as $n_e = \rho/m_p$ where $m_p$ is the rest mass of the proton.

As the initial condition, we consider a massive progenitor star that is surrounded by a dilute gas with a wind-like profile. The progenitor star is a Wolf–Rayet star with mass of ~14 solar mass at the presupernova stage, taken from model 16TI in the literature[27]. Beyond the radius of the stellar surface $R^* = 4 \times 10^{10}$ cm, we continuously connect to the external dilute gas that has a decaying power-law profile given by $\rho = 1.7 \times 10^{-14}(r/R^*)^{-2}$ g cm$^{-3}$.

We carry out three sets of simulations that consider jets with different kinetic luminosities: $L_j = 10^{49}$, $10^{50}$, and $10^{51}$ erg s$^{-1}$. In all cases, the jet is continuously injected from the inner boundary of the initial (0th) grid ($r_{in} = 10^{10}$ cm) with a half-opening angle and Lorentz factor given by $\theta_j = 5°$ and $\Gamma_i = 5$, respectively. While the initial specific heat ratio is fixed at $h_i = 100$ for the models with $L_j = 10^{49}$ and $10^{50}$ erg s$^{-1}$, a higher value $h_i = 180$ is adopted in the model with $L_j = 10^{51}$ erg s$^{-1}$. This means that the model with the highest jet power also reaches the highest terminal Lorentz factor, typically given by $\Gamma_i h_i$. We suddenly stop the steady injection at $t = 100$ s and compute the evolution until the head of the jet reaches $\sim 2 \times 10^{14}$ cm.

**Radiation transfer calculation**. Radiation transfer is calculated using a Monte–Carlo method. The method and setup are identical to our previous work[15]. By employing the output data of the hydrodynamical simulation as a background fluid, we track the trajectories of photon packets, which are an ensemble of multiple photons that have identical 4-momenta.

Initially, the photon packets are injected at the surface of a partial sphere, at a radius determined by the optical depth along the jet axis $\tau(r) = \int_r^\infty \Gamma n_e \sigma_T (1 - \beta \cos\theta_v)\mathrm{d}r'$, where $\Gamma$, $\beta$, $\sigma_T$, and $\theta_v$ are the bulk Lorentz factor, three-velocity normalized by the speed of light, Thomson cross section, and angle between the line-of-sight (LOS) of the observer and velocity direction, respectively. Here we choose a value of $\tau = 100$ for the injection radius. As shown later, our results depend only weakly on injection radius. The solid angle of the injection surface is the region with bulk Lorentz factor larger than 1.5, in order to focus on the photons from the relativistic outflow.

At the given surface, photons are injected with the intensity of black-body emission at local temperature. Intensity at a frequency $\nu$ is evaluated as $I_\nu = [\Gamma(1 - \beta \cos\theta_v)]^{-3} B_{\nu'}(T)$, where $B_{\nu'}(T) = 2h\nu'^3 c^{-2}[\exp(h\nu'/kT - 1)]^{-1}$ is the Planck function. Here $\nu' = \Gamma(1 - \beta \cos\theta_v)\nu$ is the comoving frequency, and $h$ and $k$ are the Planck constant and Boltzmann constant, respectively.

Based on the intensity, our code initially distributes numerous photon packets at the injection surface. Then the packets undergo a large number of scatterings by electrons, and are tracked until they reach the outer boundary of the calculation domain, $r \sim 2 \times 10^{14}$ cm. The distance between the scattering events is determined by drawing the corresponding optical depth $\delta\tau$. The probability for the selected optical depth to be in the range $[\tau, \tau + \delta\tau]$ is given by $\exp(-\delta\tau)\mathrm{d}\tau$. For a given optical depth, the physical distance is computed by integrating the opacity $\Gamma n_e \sigma_{KN}(1 - \beta \cos\theta_v)$ along the path of the photon over the time-evolving background fluid, where the total cross section for Compton scattering, $\sigma_{KN}$, fully takes into account the Klein–Nishina effect. At the scattering event, we first choose the four-momentum of the electron that interacts with the photon, drawn from a Maxwell distribution. Then we transform the four-momentum of the photon to the rest frame of the electron, and determine the four-momentum after the scattering based on a differential cross section for Compton scattering. Finally, we update the four-momentum of the scattered photon by transforming it to the observer frame.

**Light curves and spectral analysis**. By sampling the packets that have reached the outer boundary, we determine the properties of emission. For a given viewing angle, the light curve and spectrum are computed by collecting the packets that have propagation directions contained in a cone of half-opening angle 0.5° around the LOS. The time interval used to construct the light curve is 1 s, identical to that used in the observation to define $L_p$[1,2]. In constructing the time integrated spectra, we divide the energy range from $h\nu = 10$ eV up to 10 GeV in 100 bins equally spaced in a logarithmic scale ($\nu_n/\nu_{n-1} = 1.23$). In the current study, we consider four choices for duration of the time integration: 20, 40, 60, and 110 s. For a given duration $t_{dur}$, the spectral peak energy $E_p$ is determined by specifying the frequency at which the corresponding time integrated spectra $\nu L_\nu$ show a peak, while the peak luminosity $L_p$ is determined by identifying the maximum luminosity in the light curves within the duration. The total number of packets injected in each model is $7 \times 10^8$. This is sufficiently large to attain statistical convergence.

Since our calculation is performed in three dimensions, the jet is not axisymmetric. Hence, the emission depends not only on the viewing angle, but also on the azimuthal angle. However, the dependence is weak, and the results always reproduce the Yonetoku relation.

**On the assumption of a black-body**. One crucial ingredient that governs $E_p$ and $L_p$ is the temperature of the outflow. In determining the temperature, our simulations assume that the entire flow is characterized as a black-body. However, this prescription loses accuracy once dissipative heating takes place at regions with an optical depth $\tau \lesssim 10^5$. This is because photon production is too slow to achieve full thermal equilibrium[24]. Hence, the black-body assumption overestimates photon number density in the presence of dissipation and leads to underestimation of temperature.

In the current simulations, the black-body assumption is valid at the inner boundary $r_{in} = 10^{10}$ cm in all three models, since the optical depth is sufficiently high ($\tau \sim 10^5$). However, due to dissipative heating via the formation of shocks during propagation, the photon distribution begins to fall out of thermal equilibrium at larger radii. Nevertheless, we emphasize that error caused by shock dissipation is not large. We justify this claim below.

First, let us briefly summarize the hydrodynamical properties of the jet. Our assumption is that we inject a radiation dominant (i.e., internal energy of the radiation is larger than the rest-mass energy density) outflow which can accelerate up to a bulk Lorentz factor of a few hundreds. Since the jet is injected at $r_{in} = 10^{10}$ cm and the initial bulk Lorentz factor is 5, this means that the outflow continues to be radiation dominant at least up to the saturation radius $\sim 10^{12}$ cm for an adiabatic expansion. Note that the radiation dominant region extends to larger distances in the actual flow, since shock dissipation is present.

Shocks formed in the radiation dominant phase are considerably less efficient at heat generation (which increases the photon-to-baryon ratio $n_{ph}/n_b$ of black-body radiation) than those in the matter dominant phase[25]. As a result, our prescription does not lead to a large inaccuracy in the temperature estimation before $r \sim 10^{12}$ cm.

On the other hand, shocks can lead to some error in the temperature estimation at larger radii. However, the optical depth in this region has decreased below the value that can sustain saturated Comptonization ($\tau \lesssim 100$; unsaturated Compton zone[24]). In this region, photons cannot immediately respond to the rapid

temperature change due to dissipation. Therefore, dissipation does not have a significant effect on the resulting $E_p$ and $L_p$. Moreover, most shock heating occurs during propagation through the progenitor star, so only a small fraction of the jet matter is shock heated at these large radii.

The above qualitative discussion explains why our prescription for the temperature does not induce a large inaccuracy. Of course, further quantitative estimation is necessary to ensure that this claim is robust. For this purpose, we perform additional radiation transfer calculations which remove the assumption of a black-body. Instead we assume that, while photons at the base of the jet ($r_{in} = 10^{10}$ cm) form a black-body, the photon to baryon number ratio is conserved thereafter. With the local photon number density, the temperature is computed from $p = n_{ph}kT$. In the absence of dissipative heating, this prescription coincides with the original one. However, once dissipation begins to play a role, it leads to a larger temperature. While the original prescription corresponds to the limit of efficient photon production, this is the limit of inefficient photon production. Since the true solution should be found in between the two cases, the difference in the resulting $E_p$ and $L_p$ represents the uncertainty caused by the assumption for the temperature.

As mentioned above, the modified prescription tends to increase the temperature from the original calculation. However, rare regions with lower temperature also appear due to the entrainment of the external medium, which originally had much lower photon number density. These cases are not significant, and we again employ the black-body prescription since the lower temperature is unphysical.

Note that we also change initial condition of the thermal photons at the injection to be consistent with the modified prescription. Namely, we set the temperature and number density of the thermal photons to coincide with the updated values.

The resulting $E_p$ and $L_p$ are shown in Supplementary Fig. 1. As is apparent from a comparison with Fig. 3, no significant discrepancy is found between the two cases.

To sum up, the overall dissipative heating above the injection radius is not significant and, therefore, any discrepancy from black-body is modest. As a result, the assumption does not introduce notable error. This also implies that even if the dissipation is accompanied by possible efficient photon production, it cannot add an appreciable number of photons since the maximum number is limited to that of the black-body distribution. Thus, such effects do not modify our result either.

**On the location of photon injection**. In our simulations, photons gradually decouple and are released from the flow during expansion. Therefore, our results are insensitive to the position of injection as long as it is well below the decoupling radius. To confirm this, we perform a calculation in which photons are injected at five times larger optical depth $\tau = 500$ for the $L_j = 10^{50}$ erg s$^{-1}$ model. The results are displayed in Supplementary Fig. 2. As expected, we find no discrepancy from the original result.

Also, to demonstrate how the photons decouple from the outflow, the average comoving energy and temperature as a function of radius is shown in Supplementary Fig. 3. The photons are strongly coupled at all $\tau \gtrsim 10$, which justifies the choice of $\tau = 100$ as the injection position. This also indicates that the assumption of decoupling at $\tau \sim 50$ imposed in earlier studies[13,14] is not appropriate. As shown in the recent works[15–18] and the current study, a radiation transfer calculation is mandatory for an accurate evaluation.

**On the location of inner boundary**. Current simulations assume a somewhat large inner boundary ($r_{in} = 10^{10}$ cm) and neglect the interaction between jet and the stellar material which occurs at smaller radius. We stress, however, that it does not have a significant effect on the emission. This is mainly due to the fact that the time for the jet to reach the inner boundary from the central engine ($t_1$) is shorter than that to propagate from the boundary to the stellar surface $\sim 4 \times 10^{10}$ cm ($t_2$). Indeed, simulations[34] which employ the same progenitor model and an inner boundary ten times smaller find $t_2$ to be longer than $t_1$ by a factor $\sim 1.5$.

This implies that, even if we had started our simulation from a deeper radius, the structure of the jet formed during the initial phase ($t < t_1$) due to the direct interaction with the inner stellar matter is not reflected in the resulting emission. This is because the initial jet component would catch up to the reverse shock before the breakout and be expelled to form a cocoon, since the shock velocity is subrelativistic while the jet is relativistic. Jet material emerging later only weakly interacts with the inner material, since the initial jet component has pushed away the inner stellar material ($r < r_{in}$). However, such a signature is also washed out by the numerous shocks formed above $r_{in}$.

While the direct signature of initial interaction vanishes, it does have a modest effect on the later evolution by forming a cocoon. The cocoon's most prominent effect is confinement of jet by its pressure $P_c \sim E_c/V_c$, where $E_c$ and $V_c$ are the energy and the volume of the cocoon, respectively. The energy is proportional to the propagation time before the breakout, $E_c \sim L_j (t_1 + t_2)$. Hence, the fraction of energy that we have neglected in our calculation is estimated as $\sim t_1/(t_1 + t_2) \sim 0.4$. This means that we underestimate the cocoon pressure by 40% at most, since the true volume of the cocoon is larger. This results in a slight overestimation of the jet width[35].

It should be also noted that the cocoon is not uniform, since the sound speed at inner regions is much smaller than that at larger radii[35]. This is because the density of the star is much higher at the inner region ($\rho \propto r^{-3}$). We can roughly estimate the sound speed at the time $t_1$ as $c_s \sim (P_c/\rho)^{1/2} \sim 4 \times 10^8 (L_j/10^{50}$ erg s$^{-1})^{1/2}$ $(t_1/2.5 \text{ s})^{1/2} (r_{in}/10^{10}$ cm$)^{-3/2} (\rho/10^3 \text{ g cm}^3)^{-1/2}$ cm s$^{-1}$ ($10^3$ g cm$^3$ is the density at $r = r_{in}$). Hence, leakage time of the cocoon $r_{in}/c_s \sim 25$ s is longer than the breakout time $t_1 + t_2 \sim 6$ s which means that the inner cocoon cannot efficiently pump up the energy to larger radius. This fact further reduces the influence of early dynamics on the later evolution.

While the slight modification in the dynamics is important for addressing the detailed nature of the emission, it has no significant impact the overall properties. Even if there were a few ×10% error in the estimation of $E_p$ and $L_p$, none of our conclusions would change because such an error falls within the dispersion of the Yonetoku relation. We also note that the above issue is irrelevant for any jet material injected after the breakout, since the cocoon pressure no longer restricts the collimation.

## Data availability
The data that support the findings of this study are available from the corresponding author upon reasonable request.

## Code availability
The code used in this work is available from the corresponding authors upon reasonable request.

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

## Acknowledgements

This work was supported by JSPS KAKENHI Grant Number JP16K21630 and JP16KK0109. Numerical computations and data analysis were carried out on Cray XC30 and XC50 at Center for Computational Astrophysics, National Astronomical Observatory of Japan, the Yukawa Institute Computer Facility and Hokusai BigWaterfall system at RIKEN. This work was supported in part by a RIKEN Interdisciplinary Theoretical & Mathematical Science Program (iTHEMS) and a RIKEN pioneering project "Extreme precisions to Explore fundamental physics with Exotic particles (E3-Project)". This work was supported by NSF grant AST-1306672, DoE grant DE-SC0016369, and NASA grant 80NSSC17K0757. This work was partially supported by JSPS KAKENHI Grant Number JP16H06342 (DY), JP18H04580 (DY), and Sakigake 2018 Project of Kanazawa University (DY).

## Author contributions

H.I. and J.M. performed numerical calculations. H.I., J.M., S.N., D.C.W., M.V.B., and D.Y. discussed numerical setup and the obtained results, and worked on the text of the manuscript.
