## [Peer Review File · Nature Communications]

Reviewer #3 (Remarks to the Author):

I have read with great interest the manuscript by Ito et al. entitled "The photospheric origin of the Yonetoku relation in Gamma-Ray Bursts". I find it very well written, and containing excellent and innovative work. It definitely deserves publication. However, I do not consider it to be at the level of a Nature Communication publication. The reason for such consideration is twofold. First, the idea is not new, as the potential for photospheric emission to be able to reproduce the various ensemble correlations in prompt GRB emission has been already discussed in many papers beforehand (which are all correctly cited). The authors argue that their numerical treatment is so superior to what previously accomplished that their results are much more credible. I do not agree with such an assessment, and that's my second reason for not considering this a Nature Communications worth manuscript.

While it is true that extending the domain of the calculations makes them more credible and allows the authors to explore viewing geometries at large off-axis angles, it is also true that the increase of the domain comes at a price. The authors had to inject their jets at a somewhat large radius (10^{10} cm, for a star with radius 4×10^{10} cm) and accept a resolution that at large radii is comparable, if not larger than, R/Γ^2 . I am particularly concerned with the inner boundary. The progenitor they use is ^{16}Tl from Woosley and Heger. It has a mass of

14.15 M_{sun} at pre-supernova. The mass at radii larger than $1e10$ cm, however, is only 3 solar masses. The authors are therefore neglecting all the interaction that the remaining 10 solar masses have with the jet. Their 3D jets, at least the one shown in Figure 1, look indeed fairly smooth, with no strong shock, despite the role that the authors claim the shocks to have in the text.

In summary, I think this manuscript presents a significant advantage with respect to previous MC radiation transfer results in the size of the domain of the hydro simulation, but an equally considerable disadvantage in treating the star-jet interaction accurately and in resolving shocks at the distances at which the photosphere is located. It certainly deserves publication. Just not in a Nature journal. Even if the authors would be able to perform a higher resolution simulation injecting the jet at smaller radii, I think they should still be well aware of the limitations of a post-processing radiation models, and understand that they is not the ultimate answer. First, they do not capture adequately the jet dynamics in regions for which the optical depth is of order unity; second, they do not include particle cooling, so that an overheated region can give infinite energy to the photons rather than be cooled down as a consequence of the interaction with the radiation field.

Reviewer #4 (Remarks to the Author):

This article lays out a detailed case for photospheric emission being the dominant process in the production of GRB prompt emission, based on comparing results from numerical simulation to the

Yonetoku relation between photon peak energy E_p and peak luminosity L_p . The simulations presented do not suffer some limitations of previous works in that they are 3-dimensional; they run to a larger radius so that radiation can be calculated at a wider range of angles; and the photon spectrum and peak energy are calculated via Monte Carlo simulations of photon-electron collisions from photons injected at an optical depth of 100, rather than using an analytic model for photon peak energy. These simulations combine the best features of a number of previous approaches to produce the best hydrodynamic photospheric emission models to date. This article largely confirms the results of previous, less sophisticated approaches such as Lazzati et al. 2013, Lopez-Camara et al. 2014, Parsotan & Lazzati 2018, among many others, that show photospheric emission is consistent with the Amati, Yonetoku, and Golenetskii relations.

Comments on the article:

1. The article does a reasonably good job of address the shortcomings of it's radiation model. The section "On the validity of the numerical setup" makes the point that the total energy in photons will not be significantly impacted by dissipation so long as thermal energy is \gg than dissipation energy. However, even in this limit the number of photons produced could be an issue. For example, a large number of photons could be produced at low energy via synchrotron radiation, which could then be upscattered and impact the peak photon energy. The choice of location of photon injection should also impact the number of photons and peak photon energy.

I am not clear on why injecting photons at an optical depth of $1e5$ vs. 100 does not significantly change E_p . I would think the smaller number of photons would increase E_p by a factor of several (about 20? for injection radius of $1e10$ vs. $1e12$ cm) but I may not be understanding how photon number is calculated? Is there some mechanism for producing additional photons beyond the injection radius?

2. It may also be worthwhile to compare the method used in this paper directly with the analytic formula of Giannios 2012 (used in Lazzati et al. 2013 and subsequent works). It should be possible to calculate the same quantities (for the Yonetoku relation) with this formula for the simulation presented in this article.

I'm a bit worried that the choice of $\tau = 100$ for photon injection happens to give the right E_p . A comparison with radiation based on the Giannios formula would help. Picking a larger optical depth for photon injection, e.g. $\tau = 1000$, might also provide reassurance.

3. I am worried that the dynamic range of the hydrodynamic simulations inside the star is only 4 (1e10 to 4e10) compared to the larger range in other simulations, e.g. 40 in Lazzati et al. 2013. Although this will likely have a large effect on the particular evolution of each simulation (breakout time, energy distribution vs. angle, time to emergence of unshocked jet), it probably does not have a large impact on the correlation found in the modeled radiation. A brief comment on this limitation should be sufficient.

4. A number of important papers in the development of the photospheric model seem not to be cited. E.g. Giannios et al. 2012 and Beloborodov 2010 for the photospheric spectrum, Lazzati et al. 2009 and 2011 for hydrodynamic photosphere models, Golenetskii et al. 1983 for the Golenetskii correlation. This isn't an exhaustive list, and there may be additional papers that should be cited.

Reply to Reviewer #3

Article ID: NCOMMS-18-13818

Ito, Matsumoto, Nagataki, Warren, Barkov, and Yonetoku

" The photospheric origin of the Yonetoku relation in gamma-ray bursts"

Dear referee,

We are grateful to the referee for his/her comments. We agree that the technical issues raised by the referee are important to explore the properties of the photospheric emission in detail. On the other hand, we would like to mention that it does not have a large impact on the results that are discussed in the present study. Since the original manuscript lacked explanation on this point, we revised the paper to improve the clarity.

Below are the details of our response to the referee's comments and the description of the corresponding revisions made in the manuscript.

Comment 1

>I have read with great interest the manuscript by Ito et al. entitled "The photospheric origin
>of the Yonetoku relation in Gamma-Ray Bursts". I find it very well written, and containing
>excellent and innovative work. It definitely deserves publication. However, I do not
>consider it to be at the level of a Nature Communication publication. The reason for such
>consideration is twofold. First, the idea is not new, as the potential for photospheric
>emission to be able to reproduce the various ensemble correlations in prompt GRB
>emission has been already discussed in many papers beforehand (which are all correctly
>cited). The authors argue that their numerical treatment is so superior to what previously
>accomplished that their results are much more credible. I do not agree with such an
>assessment, and that's my second reason for not considering this a Nature
>Communications worth manuscript.

We agree that the idea of reproducing the observed correlation by the viewing angle effects is not new as mentioned in our manuscript text. The significance of the current study is on the point that our simulation has provided a robust evidence of this idea. This was possible, for the first time, by

the large dynamic range covered in three-dimension. Let us briefly summarize the important differences from the previous studies.

To our knowledge, the first calculation that has claimed to reproduce the observed correlations by the viewing angle effect was provided by Lazzati et al. 2013, ApJ, 756, 103. However, it was shown in the later studies (Ito et al. 2015, ApJ, 814, L29; Lazzati et al. 2016, ApJ, 829, 76; Parsotan & Lazzati 2018, ApJ, 853) that the evaluation of emission was oversimplified due to the imposed crude assumption. These updated studies which incorporates radiation transfer calculation enabled us to obtain more accurate quantitative estimates on the photospheric emission. On the other hand, it is also found that the considered size of the numerical domain was not sufficient to discuss the emission arising at high latitude regions ($\theta_{\text{obs}} > 4$ degree). Moreover, it is also found that imposition of 2D axisymmetry in the hydrodynamics gives rise to unphysical numerical effects around the jet axis which prevents previous work from properly estimating the on-axis emission ($\theta_{\text{obs}} \sim 0$ degree). Therefore, all previous studies could only discuss a narrow range of viewing angle and failed to explore the viewing angle dependence, which we claim is crucial for reproducing the empirical relation.

To sum up, a three-dimensional simulation that covers larger domain is necessary to adequately investigate this issue. This is exactly what we have accomplished in the current study. While the idea itself is not new, our simulation, for the first time, has confirmed that the Yonetoku relation is naturally reproduced by the viewing angle dependence.

We already pointed out the above mentioned flaws of the previous studies in the original manuscript, albeit briefly. We can explain our reservations in more detail, but since we do not want to devote many words to criticizing other works, we would prefer to keep the present form on this topic.

We agree that our calculation is not perfect, but the technical issues raised by the referee are unlikely to affect the main results and the conclusion of this paper. We describe the reasons by making responses to the subsequent comments.

Comment 2

>While it is true that extending the domain of the calculations makes them more credible
>and allows the authors to explore viewing geometries at large off-axis angles, it is also
>>true that the increase of the domain comes at a price. The authors had to inject their jets
>at a somewhat large radius (10^{10} cm, for a star with radius 4×10^{10} cm) and accept a

>resolution that at large radii is comparable, if not larger than, R/Γ^2 . I am
>particularly concerned with the inner boundary. The progenitor they use is 16TI from
>Woosley and Heger. It has a mass of 14.15 M_{sun} at pre-supernova. The mass at radii
>larger than $1e10$ cm, however, is only 3 solar masses. The authors are therefore
>neglecting all the interaction that the remaining 10 solar masses have with the jet. Their
>3D jets, at least the one shown in Figure 1, look indeed fairly smooth, with no strong
>shock, despite the role that the authors claim the shocks to have in the text.

Firstly, we would like to mention that it is not true that strong shock does not exist in our calculation. When the jet drills through the star, jet forms series of recollimation shocks (2 to 4 shocks depending on the jet power) and remains well confined due to the pressure of the cocoon. After the breakout from the stellar envelope, these shocks rapidly expand along with the jet and weaken since cocoon pressure quickly decreases. Figure 1 shows the jet configuration at this stage. While sharp structure is smoothed due to spatial resolution of the simulation at larger radii, the signatures of these shocks are not lost and reside in the flow. Also it is worth noting that innermost recollimation shock is clearly visible in the figure (slightly above the stellar surface). Indeed, the signature of these shocks is the main origin of the variability found in the lightcurves.

It is true that our simulation cannot capture very fine structure due to the limit of spatial resolution. Our spatial resolution, Δr , spans from $\sim 6e7$ to $1e11$ cm depending on radius, and the resolution around the photosphere ($r \sim 1e12$ to $\sim 1e14$ cm) is in the range of $\sim 5e9$ to $5e10$ cm. Therefore, structures that are smaller than the grid size are averaged out in the simulation, and variabilities much shorter than a timescale of a second cannot be captured. However, such short time variabilities are not important to the current study, since our main focus is on the origin of the Yonetoku relation in which the peak luminosity is defined in the time bins of a second. Thus, the finite spatial resolution in our simulation does not affect the conclusion of the manuscript. Since the above point was not clarified in the original manuscript, we added sentences that explain this issue in the subsection “Hydrodynamical simulation” of the Methods supplement (p.10, right column).

Regarding the main concern of the referee on the location of inner boundary, it is also unlikely to have a significant effect on our results. As the referee has pointed out, it is true that our calculations neglect the interaction of jet with the stellar material of 10 solar masses that is contained within $1e10$ cm. It is stressed, however, that most of the interaction which is important for the resulting emission occurs at larger radii and the impact of the earlier interactions is not significant. This is due to the fact that the time for the jet to reach the inner boundary ($r_{\text{in}} = 1e10$ cm) from the central part of the star (hereafter t_1) is shorter than that to propagate from the inner boundary to the stellar surface

$\sim 4e10$ cm (hereafter t_2). This can be explained as follows:

As shown in a number of numerical simulations (e.g., Mizuta & Ioka 2013, ApJ, 777, 162; Harrison et al. 2018, MNRAS, 477, 2128), below the stellar surface, the jet is well collimated and its head propagates with a non-relativistic speed ($\sim < 0.3 c$). The velocity gradually increases as the jet propagates, but no rapid acceleration is seen before the breakout. As a result, the propagation time is predominantly determined by the length scale for the jet to travel. Since the distance from the inner boundary to the stellar surface ($\sim 3e10$ cm) is a factor of 3 longer than that from the central part of the star to the inner boundary, $t_2 > t_1$ is expected to hold. Indeed, the simulation of Mizuta & Ioka 2013 (which employs same progenitor model and an inner boundary ten times smaller) finds t_2 to be a factor of 1.5 times longer than t_1 . Although the jet parameters are different, a similar scaling should also hold in the present case.

An important consequence of $t_2 > t_1$ is that, even if we had started our simulation from a deeper radius, the internal structure of the jet formed during the initial phase ($t < t_1$) due to direct interaction with the inner stellar gas ($r < r_{in}$) would not be reflected in the resulting emission. This is because the initial jet component would catch up the reverse shock before the breakout ($t_{br} = t_1 + t_2$) and be expelled to form a cocoon. Jet that has emerged later only weakly interacts with the inner material, since the initial jet component has pushed away the inner stellar material ($r < r_{in}$). However, such a signature is also washed out by the numerous shocks formed above r_{in} . This implies that the internal structures formed during the later interaction predominantly governs the behavior of the emission, since they can survive up to the photosphere.

The history of the initial interaction does, as mentioned above, affect the cocoon and its dynamics, which could in turn feed back to the jet and its structure. The most prominent effect of the cocoon on the dynamics is the confinement of jet by its pressure, $P_c \sim E_c / V_c$, where E_c and V_c is the energy and the volume of the cocoon, respectively. The energy deposited in the cocoon is proportional to the propagation time before the breakout ($E_c \sim L_j * t_{br}$). Hence, fraction of energy that we have neglected in our calculation is estimated as $E_{c1} / E_{ctot} \sim t_1 / (t_1 + t_2) \sim 0.4$. Since the true volume of the cocoon should be somewhat larger than that obtained in the current calculation, this implies that we underestimate the cocoon pressure by 40% at most. Therefore, the resulting jet radius ($\propto P_c^{-1/2}$), e.g. see equation 10 in Harrison et al. 2018) is maximally overestimated only by a factor of ~ 1.3 from the true value.

It should be also noted that the cocoon is not uniform as assumed in the above discussion, since the sound speed at deep regions of the star is much smaller than that at larger radii (Harrison et al. 2018).

This is because the density of the star is much denser at inner region ($\rho \propto r^{-3}$) and mixing takes place. We can roughly estimate the sound speed at the time t_1 as $c_s \sim (P_c / \rho)^{1/2} \sim 4e8 (L_j / 1e50 \text{ erg/s})^{1/2} (t_1 / 2.5s)^{1/2} (r_{in} / 1e10 \text{ cm})^{-3/2} (\rho / 1e3 \text{ g cm}^3)^{-1/2} \text{ cm/s}$. Here we assumed $V_c = r_{in}^3$ for the cocoon volume and chose $\rho = 1e3 \text{ g cm}^3$ as a fiducial value of the cocoon density which is the density of the progenitor star at $r = r_{in}$. Hence, leakage time of the cocoon $r_{in} / c_s \sim 25 \text{ s}$ is longer than the breakout time $t_1 + t_2 \sim 6 \text{ s}$ which means that the inner cocoon cannot efficiently pump up the energy to larger radius. This fact further reduces the influence of early dynamics on the later evolution.

While the slight modification in the dynamics might be important for addressing the detailed properties of the emission, it is highly unlikely that it will significantly modify the overall properties (i.e., E_p and L_p) explored in the present study. The dispersion in the Yonetoku relation is larger than the maximum possible error we describe above. It is also worth noting that the above issue is almost irrelevant for the dynamics of jet material that is injected after the breakout (and hence the late-time emission), since the cocoon pressure no longer restricts the collimation of jet.

To sum up, the relatively large inner radius imposed in our calculation does not induce serious ambiguity in our results and conclusions. While we agree it is important to start from smaller inner radius is preferred to gain deeper insight of the emission, recalculation with smaller inner boundary cannot be accomplished within a reasonable time. Therefore, we would like to focus on the current results in this manuscript. On the other hand, since the above issue was not clarified in the original manuscript, we have added additional sentences on this topic in the subsection ‘‘On the location of inner boundary’’ of Method supplement to improve the clarity.

Comment 3

> In summary, I think this manuscript presents a significant advantage with respect to previous
 > MC radiation transfer results in the size of the domain of the hydro simulation, but an equally
 > considerable disadvantage in treating the starjet interaction accurately and in resolving shocks
 > at the distances at which the photosphere is located. It certainly deserves publication. Just not
 > in a Nature journal. Even if the authors would be able to perform a higher resolution simulation
 > injecting the jet at smaller radii, I think they should still be well aware of the limitations of a
 > postprocessing radiation models, and understand that they is not the ultimate answer. First
 > they do not capture adequately the jet dynamics in regions for which the optical depth is of
 > order unity; second, they do not include particle cooling, so that an overheated region can give
 > infinite energy to the photons rather than be cooled down as a consequence of the interaction

>with the radiation field.

The limitation of resolution and large injection does not have significant effect on our conclusions for the reasons mentioned above in the response for Comment 2.

We are aware of the further limitation of our postprocessing calculation raised by the referee, but these ingredients also do not change our conclusions. Regarding the radiative feedback on the dynamics, the effect plays an important role when the radiation energy is comparable to or larger than that of the matter in regions where optical depth is close to unity. However, in the current simulation, the matter energy is always dominant at such regions. The highest energy ratio of the photon to the total (sum of photon and matter) is realized near the on-axis region and is $\sim 30\%$ at most. The ratio decreases at higher latitude regions and drops below few % at $\theta \gtrsim 4$ degree. This fact implies that the radiation feedback has only modest influence around the on-axis region and can be neglected at high latitude regions. Also it is important to recognize that the radiation feedback cannot modify the global geometry of the outflow even at the on-axis region. This is because, since both radiation and matter moves outward with almost speed of light, the radial diffusion relative to the matter is on a scale $\sim r / \Gamma^2 \sim 4e7 (r / 1e13 \text{ cm}) (\Gamma / 500)^2 \text{ cm}$, and also lateral diffusion is limited to be within an angle $\sim 1 / \Gamma \sim 0.11$ degree. Thus, while the radiative force can work to slightly smoothen the structures contained within this small special scale, it cannot modify the largescale morphology ($dr \sim \text{few } 1e10 \text{ cm}$, $d\theta \sim \text{few degree}$) that is discussed in the current manuscript.

Regarding the effect of particle cooling due to the interaction with radiation, the second issue raised by the referee, it also does not change the essential features found in the current study. While we agree that the temperature of the matter has some ambiguity, we do not agree that an overheated region appears in the current simulation. Actually, the situation is opposite. Our simulation tends to underestimate the temperature of the plasma due to the assumption of full thermal equilibrium between matter and radiation ($P=aT^4$). Therefore, true temperature can be higher at regions with optical depth less than $\sim 1e5$ in which dissipations can give rise to departure from thermal equilibrium. It is emphasized, however, that the effect of this departure is not large in the current simulations and do not give rise to significant change in the results. This is confirmed quantitatively in the subsection entitled “On the validity of the numerical setup” in the Methods supplement.

For the above reasons, we again emphasize that, while the effect raised by the referee is important for capturing the full physics of the emission, it cannot provide a significant modifications to the conclusions of the manuscript, which are robust even when few times 10 % errors are present in the

evaluation of E_p and L_p .

All the major revisions made in the manuscript are written in red. We hope that the referee will agree that our revised manuscript is now acceptable for publication.

Sincerely,

H. Ito

J. Matsumoto

S. Nagataki

D. Warren

M. Barkov

D. Yonetoku

Reply to Reviewer #4

Article ID: NCOMMS-18-13818

Ito, Matsumoto, Nagataki, Warren, Barkov, and Yonetoku

" The photospheric origin of the Yonetoku relation in gamma-ray bursts"

Dear referee,

We are grateful to the referee for comments that have helped us to improve our paper. We agree that the issues raised by the referee are important. Therefore, we have revised our manuscript taking into account all the comments.

Below are the details of our response to the referee's comments and the descriptions of the corresponding revisions made in the manuscript.

Comment 1

>The article does a reasonably good job of address the shortcomings of it's radiation model.

>The section "On the validity of the numerical setup" makes the point that the total energy in

>photons will not be significantly impacted by dissipation so long as thermal energy is \gg than

>dissipation energy. However, even in this limit the number of photons produced could be an

>issue. For example, a large number of photons could be produced at low energy via

>synchrotron radiation, which could then be upscattered and impact the peak photon energy.

>The choice of location of photon injection should also impact the number of photons and peak

>photon energy.

>I am not clear on why injecting photons at an optical depth of $1e5$ vs. 100 does not significantly

>change E_p . I would think the smaller number of photons would increase E_p by a factor of

>several (about $20?$ for injection radius of $1e10$ vs. $1e12$ cm) but I may not be understanding

>how photon number is calculated? Is there some mechanism for producing additional photons

>beyond the injection radius?

We agree with the referee that dissipation can cause a notable impact on the spectral peak energy, if

the accompanied photon production is efficient enough to outnumber the advected photons. This is true even when the dissipated energy is smaller than that of the thermal energy. In such a case, however, photon distribution prior to the dissipation must be far from the equilibrium (i.e., photon number much smaller than that of the blackbody distribution) to significantly increase the photon number. This is because the maximum number of photons is limited by the blackbody distribution which does not differ largely by the moderate dissipative heating. In other words, even if there were efficient photon generation, it quickly balances with absorption processes after adding up small number of photons in the system. Thus, if the photon distribution before the dissipation is close to the blackbody and the dissipation energy is not significant compared to that of the initial thermal energy, the impact of photon generation is small.

This is one of the points that we intended to describe in the section “On the validity of the numerical setup”. In our simulation, the inner boundary of the calculation domain is located at the region with optical depth of $\tau \sim 1e5$, at which full thermal equilibrium (blackbody) is likely to hold. In this section, we have shown that subsequent dissipative heating is not significant throughout the flow. This implies that even if efficient photon production occurs with the dissipation, it cannot significantly increase the photon number and, therefore, does not affect the spectral peak energy. Since the above issue of photon production was not explicitly mentioned in the original manuscript, we added new paragraph at the end of the subsection “On the assumption of black body” in the Methods supplement to improve the clarity.

The above point is the reason for why injecting photons at an optical depth of $1e5$ vs. 100 does not significantly change E_p . At regions with optical depth smaller than $\sim 1e5$, any possible photon production cannot increase the photon number appreciably. This means that photon number per baryon is nearly conserved throughout the flow and is close to that of the blackbody with a local temperature calculated as $P = aT^4$. Therefore, the results do not significantly depend on the injection radius.

Comment 2

> It may also be worthwhile to compare the method used in this paper directly with the analytic
> formula of Giannios 2012 (used in Lazzati et al. 2013 and subsequent works). It should be
> possible to calculate the same quantities (for the Yonetoku relation) with this formula for the
> simulation presented in this article.

>I'm a bit worried that the choice of $\tau = 100$ for photon injection happens to give the right E_p .
>A comparison with radiation based on the Giannios formula would help. Picking a larger optical
>depth for photon injection, e.g. $\tau = 1000$, might also provide reassurance.

We agree that comparison with result obtained by employing the analytical formula of Giannios 2012 is worth to be checked. However, applying the formula directly to the hydrodynamical simulation (as done in Lazzati et al. 2013 and subsequent studies) requires new code development which cannot be done in a short time scale. Since this is not strongly physically motivated, instead, we show how the photons are decoupled from the outflow and form the observed E_p in the subsection “On the location of photon injection” of the Methods supplement. In the newly added Supplementary Figure. 3, it is clearly seen that at an optical depth $\tau \sim 50$, where Lazzati et al.2013 assumes as the decoupling radius (photosphere) based on the study Giannios 2012, the photons are still strongly coupled with the flow. The peak energy as well as the luminosity at the position systematically larger than those of the observed ones, and the difference can be as large as order of magnitude. This clearly indicates that application of Giannios formula is not appropriate. This is not a surprising result, since the formula is derived under the assumption of steady spherical flow which are subject to continuous dissipative heating, while the current simulation computes unsteady multi-dimensional flow in which dissipation takes place only locally at shocks.

To further check the validity of the choice of $\tau=100$ for the injection, we have calculated the case of injection at 5 times larger optical depth $\tau=500$. This new simulation confirmed that the results do not vary with the choice of larger optical depth. The result is shown in Supplementary Figure 2 and described in the Methods supplement.

Comment 3

> I am worried that the dynamic range of the hydrodynamic simulations inside the star is only 4
>(1e10 to 4e10) compared to the larger range in other simulations, e.g. 40 in Lazzati et al. 2013.
> Although this will likely have a large effect on the particular evolution of each simulation
>(breakout time, energy distribution vs. angle, time to emergence of unshocked jet), it probably
>does not have a large impact on the correlation found in the modeled radiation. A brief
>comment on this limitation should be sufficient.

We agree with the referee that our choice of the inner boundary position has a certain effect on the hydrodynamical evolution and, therefore, on the emission properties. Since we have employed relatively large value for the injection radius, we neglect some portion of the initial interaction between the jet and progenitor star. However, as the referee mentions, this fact is unlikely to induce a crucial error on the overall properties of the emission (i.e., E_p and L_p) discussed in the current study. Since this issue was not clarified in the original manuscript, we have added new subsection “On the location of inner boundary” in the Methods supplement that explains the possible impact of the inner boundary position to improve the clarity of the paper.

Comment 4

> A number of important papers in the development of the photospheric model seem not to be
> cited. E.g. Giannios et al. 2012 and Beloborodov 2010 for the photospheric spectrum, Lazzati
> et al. 2009 and 2011 for hydrodynamic photosphere models, Golenetskii et al. 1983 for the
> Golenetskii correlation. This isn't an exhaustive list, and there may be additional papers that
> should be cited.

We agree with the referee that we have not cited some important papers in the original manuscript. Therefore, in the revised manuscript, we have cited the papers that are suggested by the referee and also several papers in addition that we believe to be important.

All the major revisions made in the manuscript are written in red. We hope that the referee will agree that our revised manuscript is now acceptable for publication.

Sincerely,
H. Ito
J. Matsumoto
S. Nagataki
D. Warren
M. Barkov
D. Yonetoku

REVIEWERS' COMMENTS:

Reviewer #3 (Remarks to the Author):

I have read the authors reply and the revised manuscript. I do not have any more technical comments or suggestions. It is a sound and well-written manuscript. However, I still do not think the results are new and impactful enough to warrant publication in Nature Communications.